# Eco-Dimensioning Approach for Planar Transformer in a Dual Active Bridge (DAB) Application

**Glauber de Freitas Lima** [1,*] **, Boubakr Rahmani** [1,2] **, Maud Rio** [2] **, Yves Lembeye** [1] **and Jean-Christophe Crebier** [1]

[1] G2Elab, Grenoble INP, CNRS, University Grenoble Alpes, F-38000 Grenoble, France; boubakr.rahmani@grenoble-inp.fr (B.R.); Yves.Lembeye@g2elab.grenoble-inp.fr (Y.L.); jean-christophe.crebier@g2elab.grenoble-inp.fr (J.-C.C.)

[2] GSCOP, Grenoble INP, CNRS, University Grenoble Alpes, F-38000 Grenoble, France; maud.rio@g-scop.eu

[*] Correspondence: glauber.de-freitas-lima@g2elab.grenoble-inp.fr; Tel.: +33-(0)-74916-5697

**Abstract:** Power electronics converters are traditionally designed regarding efficiency, power density, cost, and reliability figures of merit. Today, with the extreme spread of power electronic applications in our modern societies, together with the earth limits in terms of materials resources, it is important to consider the ecological impact of the converter not only during its usage, but over its whole life cycle. This article introduces an eco-dimensioning methodology for analyzing and accounting for the energy consumption over the entire converter life. The analysis is applied on a small DC-DC converter considering the main components dual active bridge (DAB) converter. The planar transform is one of the key elements modeled in this article, including material and manufacturing conditions. The traditional and eco-dimensioning approaches are carried out and compared in order to emphasize the possible consequences on total energy cost.

**Keywords:** planar transformer; eco-design; environmental impact evaluation; product efficiency; DC-DC converter; dual active bridge (DAB); life cycle optimization





## 1. Power Electronic Design

### 1.1. Conventional (Efficiency vs. Power Density Focused)

Up to now, to achieve cost reductions due to the growing need for information and telecommunication technologies with the development of power electronic systems, it has been popular to increase the power density (up to 35 kW/dm$^3$ in a DC-DC converter [1]). Such an achievement is mainly performed by means of switching frequency increase and the use of forced cooling systems with the drawback of global efficiency reduction (<95%).

According to [2,3], nowadays, 40% of the total used energy is enabled by electric power, and this figure could reach 50% by 2030 [3] and 60% [2] by 2040. Such important growth implies that the role of power electronics is not only power density focused, but rather related to efficiently and ecologically processing the energy required by different specific applications. The applications can vary from small power (W) such as mobile phone chargers to higher power applications such as electric cars (kW), small and large renewable power plants, trains, metros (MW), and DC transmission lines (GW), etc.

Nowadays, with the advance in technologies for semiconductors (increase in junction temperature, ON/OFF states and rise/fall time improvement and optimization) and magnetic materials (higher operating temperatures, reduction of hysteresis losses) as well as by allowing power density reduction [2] (e.g., 2.2 kW/dm$^3$ [4], 7kW/dm$^3$ [3]), design with so-called extreme efficiency converters [5] (e.g., 98% [4], 99.2% [3,6]) have come to represent the major success of power electronics realization.

With the use of active and passive components as well as auxiliary circuity and cooling systems (when needed) [1], power electronic system design is conceived to process electrical specifications of desired voltages ($V_i$, $V_o$, $\Delta V_o$), currents ($I_o$, $\Delta I_o$), and power ($P_o$)

following the mandatory and safe standards (e.g., EMI (electro-magnetic interference), safety standards among others).

To proceed with the traditional power electronic design, a multi-objective optimization [3,6–8] is performed to guarantee optimum performances through Pareto fronts (e.g., Efficiency × Power Densities (W/dm$^3$ and W/kg) × Costs (€/W) × MTBF (Mean Time Between Failure) (h) $\eta$–$\rho_{box}$–$\sigma_P$ [3,8]). Figure 1 presents a conventional multi-objective flowchart based on [7] for applying and designing any power converter.

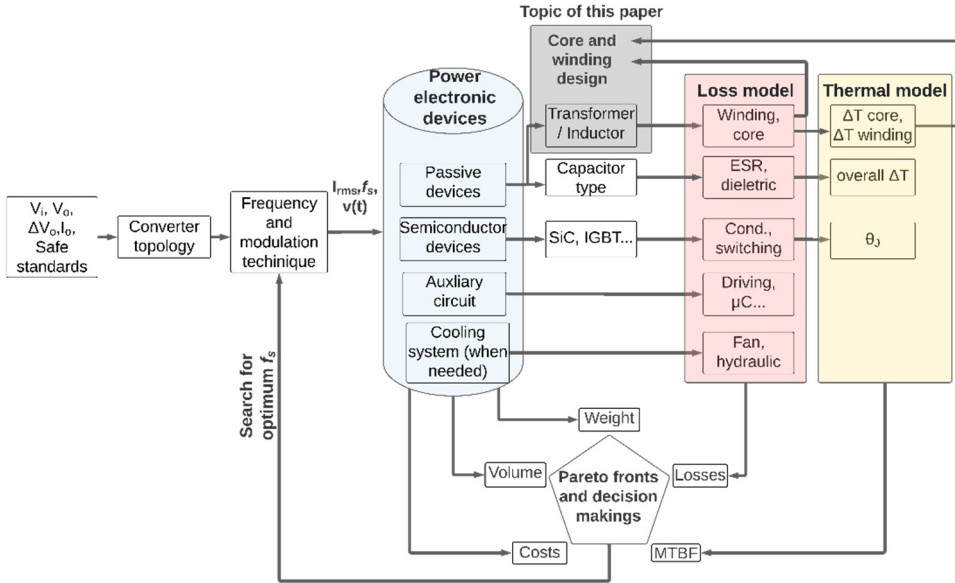

**Figure 1.** Summarized flowchart of a conventional design with global optimization of the main needs in a power electronic systems.

As can be seen, conventionally, power electronic choices are based on many criteria, such as failure rate, volume, weight, power losses, and costs. When focusing on this multi-objective conventional approach, extremely highly efficient (~96–99%) and/or power density systems (~2 kW/dm$^3$–~6 kW/dm$^3$) are conceived [3,7,8] in a Pareto front behavior over a wide load charge profile.

As covered in [3,7], designing a very efficient system (over 90%) while assuring high power density is a challenging work, as there is always a compromise between peak efficiency and power over total volume. The reason for this cognitive dissonance regarding "efficiency vs. power density", linking the need for miniaturization and maintenance of high efficiency, is mainly related to the switching frequency ($f_s$) in which power converters operate. By designing and operating at high frequencies, e.g., 300 kHz and above, the passive elements tend to decrease in volume and material, making the prototype more compact. Meanwhile, the increase of the $f_s$ may impose an increase of the power losses all over the converter (switching, skin effect, magnetic hysteresis losses, EMI filter effort), as well as increasing the design complexity (driving circuits' interconnects, EMI compliance, dynamic response management, µC bandwidth).

Therefore, such an important choice of $f_s$ is one of the main design parameters which justifies the design of the close loop introduced in Figure 1. Such a parameter will determine the power losses and the volume, which is directly related to the global life cycle energy cost, as covered in Section 1.2.

### 1.2. Eco-Design

When considering eco-design, it is necessary to represent and to take into account complementary parameters, such as the entire energy dissipation over the converter life cycle from the material extraction, and refine them to usages and finally end-of-life management. While the traditional design mainly focuses on optimizing the converter for its usage, looking after best efficiencies, the eco-design also considers the two other life

stages in order to prevent the converter from only being optimized for its usage, leaving for example a fully non-recyclable waste at the end of life or relying on extremely polluting materials. Today, with the pressure that our modern societies put on earth materials and energies, it is urgent to eco-design power converters.

In Figure 2, the complementary design blocks to the schematic introduced in Figure 1 are proposed to account for eco-dimensioning. In such a way, the weight, volume, efficiency, and MTBF are used to feed another set of models seeking to evaluate the total energy losses, the design model functionalities, the potential recycling fraction, and the waste and pollution generated during the life cycle.

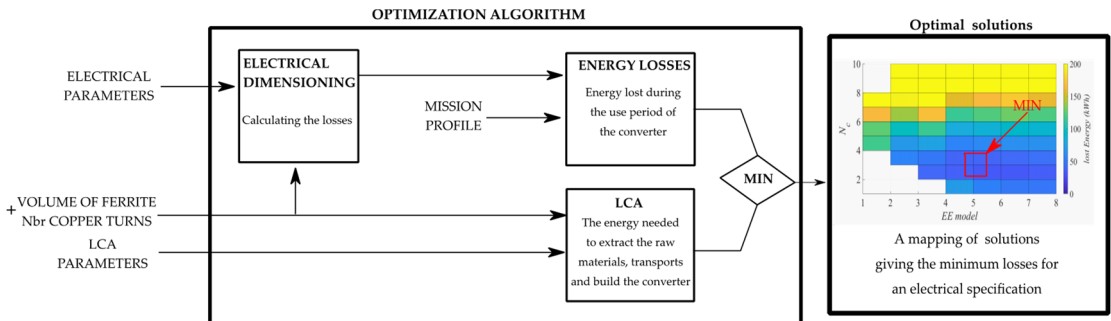

**Figure 2.** Summarized flowchart of a conventional design mixed with a Life Cycle Assessment vision (LCA) for global optimization of the main needs in a power electronic systems.

In the traditional concept, the weight, volume, and efficiency are chosen according to subjective specifications, e.g., costumer needs and available space constraints, whereas in eco-design, they have a direct impact on the objective.

In such a way, if it is easy for everyone to link converter efficiency and usage energy losses. An important difference can be related to the efficiency to be optimized. Whereas the peak efficiency could be optimized to outline a key characteristic on the converter, in eco-design, the mission profile and operating conditions are strongly impacting the energy losses under usage, bringing the design to consider not only the peak efficiency or the efficiency at maximum power efficiency over a wider power and operating condition range.

Similarly, regarding the active and passive components, and moreover their materials, the optimization of the component losses and power density may lead to the use of "exotic" materials, i.e., difficult to produce. This optimization process could be challenged by the environmental impact of the materials and/or technologies used to manufacture them.

One very tricky point is related to the integration process which receives a large plebiscite, increasing at best power densities, as well as performances under high switching frequencies thanks to reduced interconnects. The integration brings, on one side, collective processes, higher reliability, and cost reduction, but on the other side, it produces very heterogeneous systems that are difficult to disassemble and recycle. The processes used to extract the various materials are usually very polluting and energy consuming and most of the time only part of the materials remains fully useful, with the others being at best down-cycled. In this context, the life cycle assessment (ISO 14040:2006) of the system designed enables to model alternative scenarios to identify the potential environmental impact sources, support decision-making during the design process, and converge to an present an efficient life cycle system.

## 2. Dual Active Bridge (DAB)

There are different classes and topologies of power converters throughout industry and literature, e.g., Phase-shift DC-DC converters, Resonant DC-DC converters, DC link AC-AC converters, Matrix AC-AC converters, etc. Regarding topology, its selection is based on criteria such as efficiency, profile charge characteristics, power density, galvanic isolation, power directionality, costs, dynamic behavior and controllability, expertise, and complexity, etc. The scope of this section does not intend to impose the most optimum

topology choice, but rather describe one of the most popular topologies, the DAB converter, as a representative user case. This will be considered using single-phase-shift (SPS) modulation characteristics [9,10] and benefits found in our laboratory in previous research works [9,11,12].

The DAB has become a popular DC-DC converter over the last ten years that allows galvanic isolation, bidirectionality in power, symmetricity, controllable output, high power density, and efficiency. Due to such exceptional performance and features, it is mostly used in renewable energy systems [13,14], energy storage systems [15], DC distribution systems [16], and solid-state transformer (SST) [17]. The topology is presented in Figure 3.

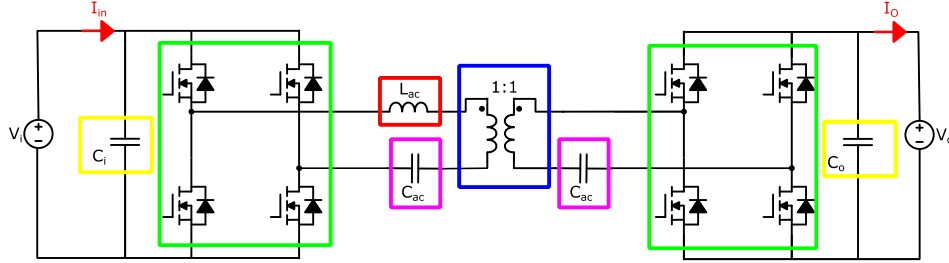

**Figure 3.** Dual Active Bridge (DAB) topology used as scenario study. Colored scares are highlighting the main components' types.

Even if ZVS (zero voltage switching) is a driving feature to reduce switching losses, the choice and optimization of the switching frequency remains a key parameter in the DAB, impacting on the design of the transformer (highlighted in blue), the choice of the AC inductance (highlighted in red), the AC capacitor (highlighted in purple), the active devices, here selected as semiconductor MOSFETs (metal-oxide-semiconductor field-effect transistor) (highlighted in green), and finally, the input and output capacitor (highlighted in yellow).

The DAB allows the implementation of different modulation techniques [18–21] that increase ZVS and ZCS (zero current switching) range, decrease RMS (root mean square) current value, and therefore improve efficiency and power density throughout the charge profile of a given application. In this work, the ordinary SPS modulation [9,10] was chosen due to its simplicity and due to an unpretentious profile charge that allows good performance: a constant input voltage $V_i$ = 20 V, a constant output voltage $V_o$ = 20 V and an output current $I_o$ of 2 A during half of the time and 1 A during the other half.

As in [9], in order to have a comprehensive analysis, the important parameters of design are portrayed in a dimensionless and meaningful plane, as a function of the static gain $M$ versus a normalized output current $\gamma$, given by (1) and (2), respectively.

$$M = \frac{V_o}{V_i} \tag{1}$$

$$\gamma = \frac{2 f_s L_{ac} I_o}{V_i} \tag{2}$$

Figure 4 presents the normalized RMS current (3) with respect to the output current in the parametrized output plane including the ZVS regions

$$\bar{I}_{rms} = \frac{I_{rms}}{I_o} \tag{3}$$

It is possible to conclude from Figure 4 that a safe design guarantying ZVS is located for the design criteria $\gamma_{min}$ = 0.05 and $\gamma_{max}$ = 0.1. This will allow, for example, minimization of conducting losses up to 5% with a circulating current value $I_{rms}$ of around 15% larger than the maximum output current $I_o$.

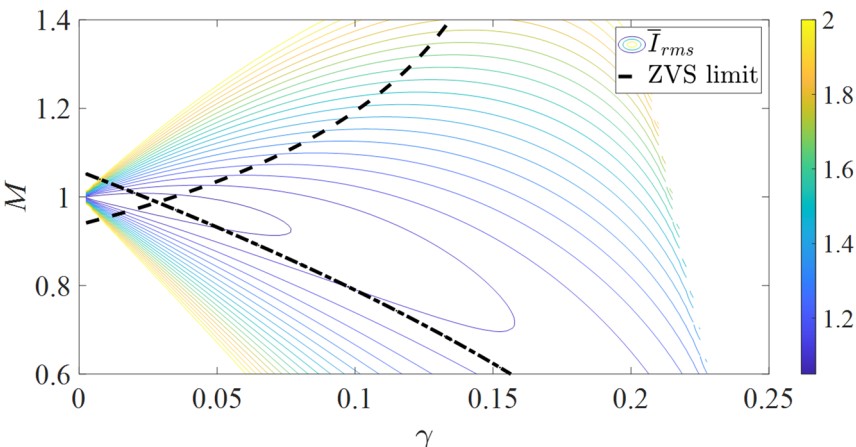

**Figure 4.** Normalized RMS current mapping of DAB under single phase-shift modulation including ZVS regions.

As it is possible to notice in Figure 5, all the devices are selected in a loop together with the choice of the switching frequency, which is an output of the minimizing function result. In this work, the selection of MOSFET, capacitors and inductor family are already given from previous work, which is presented in Table 1. Once they are selected, their losses must be computed and analyzed.

**Table 1.** Devices or family used for loss analysis.

| Main Devices | | |
|---|---|---|
| MOSFET | SiSA10DN, N-Channel 30 V | 5 mΩ; Qg = 18.75 nC |
| Inductor $L_{ac}$ | Coilcraft XEL6060 | |
| Capacitors | Ceramic | $R_{ac}$ = 2 mΩ |
| Transformer cores | EE Planar Ferroxcube [22] | Section 3. |

For the MOSFET, as the ZVS is assumed, the only losses considered are due to $Q_g$ and the conducting losses as presented in (4) thanks to datasheets. $R_{mosfet}$ can be expressed with respect to the junction temperature estimate and gate voltage, while $Q_g$ and $V_i$ are respectively derived from operating conditions.

$$P_{Mosfet} = R_{mosfet}I_{rms}{}^2 + Q_g V_i f_s \qquad (4)$$

The losses of the $C_{ac}$ are simply computed as in:

$$P_{Ca} = R_{Ca}I_{rms}{}^2 \qquad (5)$$

Then, the value of the inductance is chosen according to the load profile to respect ZVS condition and maximize efficiency, as already discussed and presented in Figure 4. By considering $\gamma_{max}$ = 0.1, the value of the inductance should go with the switching frequency as presented in (6).

$$f_s L_{ac} = \frac{V_i \gamma_{max}}{2 I_{o_{max}}} = 0.5 \Rightarrow L_{ac} = \frac{0.5}{f_s} \qquad (6)$$

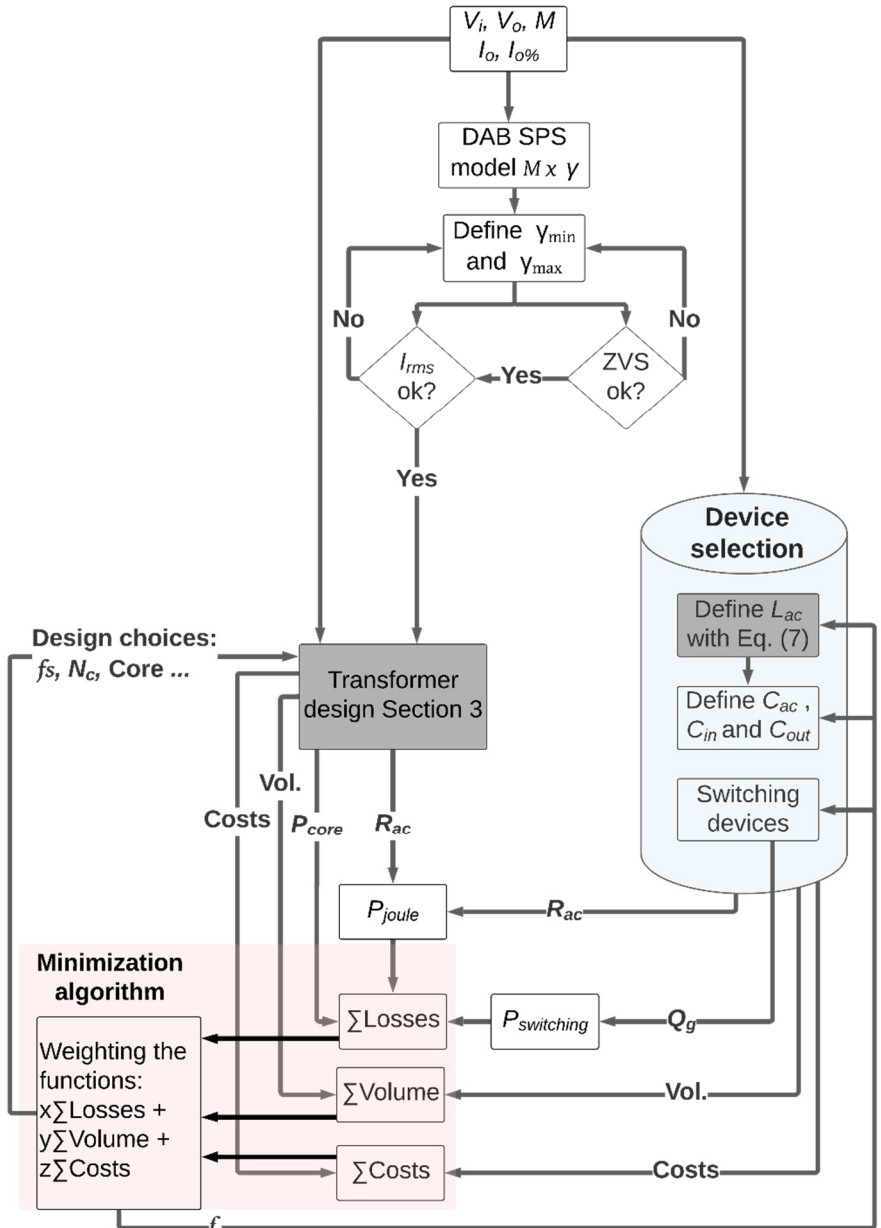

**Figure 5.** Optimizing routine for minimization of losses, volume or costs considering the main aspects in a DAB with SPS modulation.

Finally, considering current ratings, size, cost, and AC losses, the choice of the inductor can be selected from manufacturers, e.g., Coil Craft XEL6060. To take into account the DC resistance present in the inductors with respect to the inductance value, a linearization of its $R_{dc}$ vs. $L_{ac}$ is performed, as shown in Figure 6. The linear function is presented in (7).

$$R_{Lac_{dc}} = 2.9 \times 10^3 \times L_{ac} + 0.00024 \tag{7}$$

Following from its spice model given by the manufacturer, the series AC resistance can be expressed as in (8).

$$R_{Lac_{ac}}(f_s) = R_{Lac_{dc}} + 10^{-6}\sqrt{f_s} \tag{8}$$

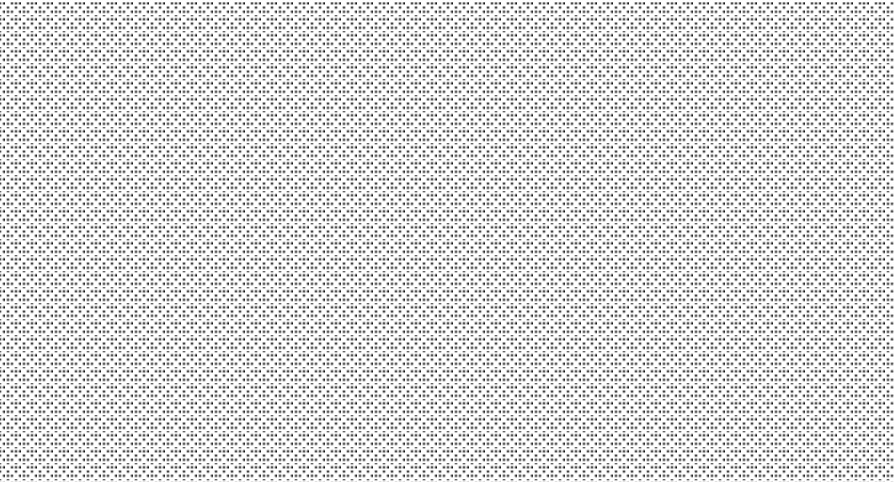

**Figure 6.** Linearization of the DC resistance with respect to inductance found on Coilcraft's XEL6060 series.

Finally, substituting (8) in (7), and imposing the condition as in (6), the value of the AC resistance can be estimated with respect to the switching frequency in (9). The result is presented in Figure 7. Notice that this result depends on the technology of the inductor, that is, its DC and AC performances.

$$R_{Lac_{ac}}\left(L_{ac} = \frac{0.5}{f_s}\right) = \left(2.9 * 10^3 \frac{0.5}{f_s} + 0.00024\right) + 10^{-6}\sqrt{f_s} \tag{9}$$

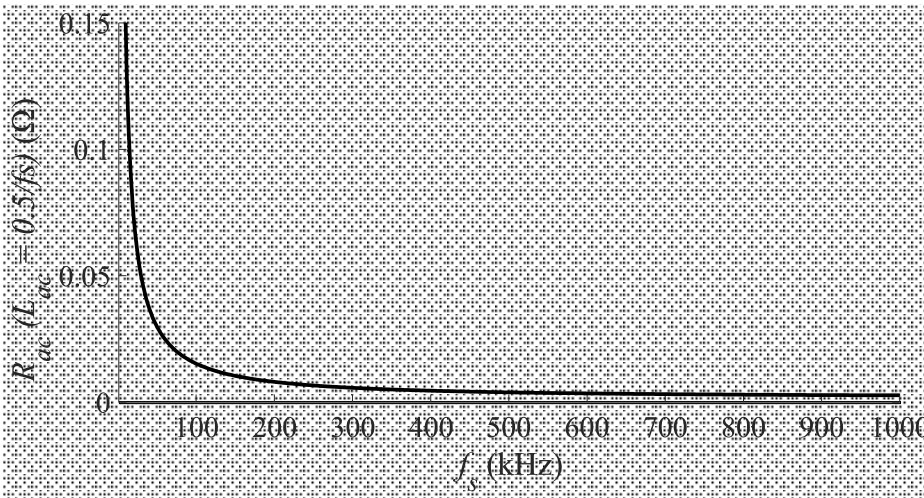

**Figure 7.** Estimated AC resistance considering the XEL 6060 series from Coil Craft by imposing $L_{ac} = 0.5/f_s$.

Then, after summing the losses on the MOSFETs, $C_{ac}$, and $L_{ac}$, the results are plotted with respect to the switching frequency as presented in Figure 8. Notice that an optimum switching frequency is found around 100 kHz. However, the losses on the transformer will be presented and included in Section 3.

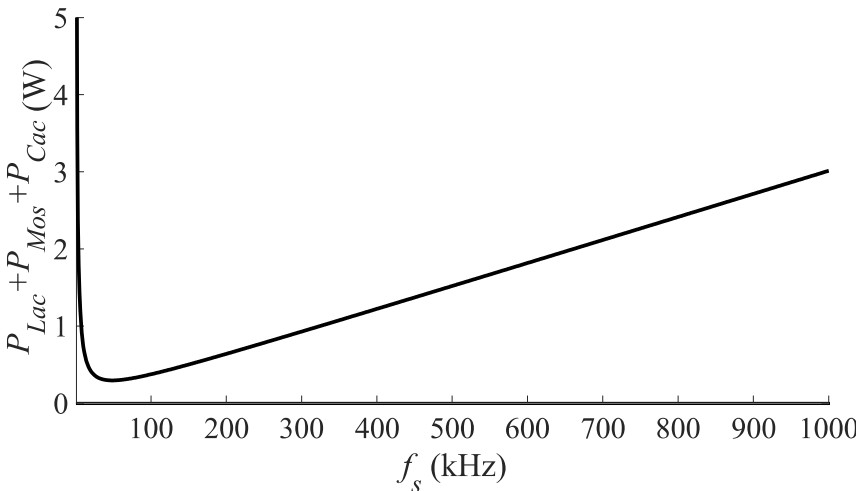

**Figure 8.** Estimated total losses with respect to switching frequency fs regardless the transformer.

## 3. Planar Transformer

In complement to the extra losses, the transformer plays a key role when it comes to weight and volume [23] in the design of the DAB. Even though loss sharing is not as significant, in an optimized design presented in [8], it occupies around 41% of the volume shares. In the work presented here, its volume represents about 45%, while its mass around 80% in the existing hardware prototype available in the laboratory from previous work [11,12]. In [10], a methodology is presented for designing a high frequency wire wound transformer for a DAB with SPS modulation 2.2 kW, by previously fixing one switching frequency design at 40 kHz. Therefore, the figure of merit of design is not clear with respect to power density and efficiency. The advantage of planar transformers [24,25] when compared to wire wound transformers is good thermal characteristics (due to higher surface area), easy assembly in power converters, low profile (e.g., by 37.3% as well as loss reduction [26]), predictability, repeatability, and cost reduction under certain conditions. Some disadvantages, however, include a larger footprint and higher winding capacitances.

Two different winding technologies may be used when prototyping a planar transformer: one made of PCB (printed circuit board), and another one, named copper foil. In Figure 9, the main aspects concerning the PCB winding technology are presented, which will be used through this work for result evaluation.

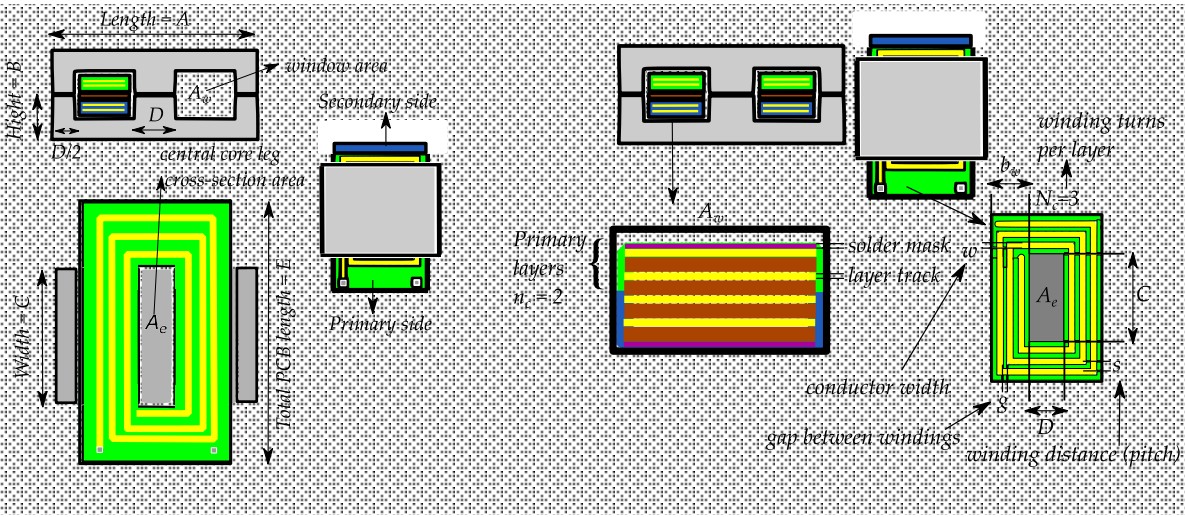

**Figure 9.** Important aspects in a EE planar transformer: (**a**) Core geometry; (**b**) Winding aspects (example of a 6:6 planar transformer).

### 3.1. Classical Transformer Design

A classical equation, used throughout industry and academia for choosing in a practical way the right ferrite core is based on the area product [27], which is the product between the window area $A_w$ and the magnetic cross section area $A_e$: $A_e A_w$ for forward converter types, as shown in (10). Such an equation allows to easily find the core provided on the manufacturer datasheets.

$$A_e A_{w_{\min}} = \frac{S_{max}}{K_r K_f J_{max} B_{peak} f_s} \tag{10}$$

where:

- $S_{max}$ is the total apparent power
- $B_{peak}$ is the magnetic induction (T), usually saturates at 0.3~0.4 T, but is often limited by the power losses;
- $J_{max}$ is the current density (A/cm$^2$)
- $K_r$ is the fulfill factor = 0.1~0.2 defined by the winding technology and isolation needs. As in [24], it is adopted 0.14 and can be later verified at the end of the design;
- $K_f$ is the form factor defined by the waveform in the input of the transformer (equal to 4 due to DAB a square waveform and 4.44 for sinusoidal one);

Notice that, depending on the choice of the current density $J_{max}$, magnetic induction $B_{max}$, and switching frequency $f_s$, different area products can be chosen, predefining the geometry and volume of the transformer, which are important parameters for efficiency, and power density. To calculate the efficiency, two losses occur inside the transformer, namely core and copper losses.

Normally, the core losses are calculated using the Steinmetz method, in which the manufacturer provides empirical parameters to be used in the Steinmetz equation adapted for a square waveform with 50% duty cycle commonly found on full-bridge DC-DC converter [28]. In this work, the employed parameters α, β, and $c$ of a 3F3 ferrite were provided by the manufacturer, FerroxCube [22], and are mostly dependent on the core temperature and $f_s$. Commonly, for ferrite materials, lower temperatures imply higher losses, and for guaranteeing high efficiency, a temperature of 25 °C [23] was used throughout this equation. The total core loss is then presented in (12) by multiplying by the core volume $V_e$.

$$P_{core/V_e} = 2^{2\alpha-1} 0.5^{\beta-\alpha+1} c f_s{}^\alpha B_{peak}{}^\beta \tag{11}$$

$$P_{coreTotal} = P_{cor/Ve} \times V_e \ [\text{W}] \tag{12}$$

Regarding copper losses, the eddy losses are calculated based on Dowell factor and the general field solutions [25] defined in (13) and (14).

$$P_{winding} = 2 I_{RMS}{}^2 R_{dc} F_r \, [\text{W}] \tag{13}$$

$$F_r = \frac{R_{ac}}{R_{dc}} = \frac{\xi}{2} \left[ \frac{\sinh\xi + \sin\xi}{\cosh\xi - \cos\xi} + (2m-1)^2 \frac{\sinh\xi - \sin\xi}{\cosh\xi + \cos\xi} \right] \tag{14}$$

where $\xi$ is the ratio $h/\delta$, $\delta$ is the skin effect defined in (15), and $m$ is the ratio defined by the MMFs at the limits of the layers in which will depend on the interleaving arrangement technique. The value of $m = 1$ is the choice for this work.

$$\delta = \sqrt{\frac{\rho_{cu}}{\pi f_s \mu}} \tag{15}$$

The DC resistance basic equation is presented in (16). However, besides depending on the number of turns and core geometry, it will also depend on the winding technology. For example, in Tables 2 and 3, the presented design guide and rules for PCB fabrication are from Wurth Elektronik [29]. The choice of them will impact on the efficiency as well as manufacturing costs.

**Table 2.** PCB main guideline and rules available in [29].

| Outer layers/Inner Layers Spacing (µm) | Min. Finished Copper (µm) ($h_c$ = 18 µm) | Min. Finished Copper (µm) ($h_c$ = 35 µm) | Min. Finished Copper (µm) ($h_c$ = 70 µm) | Min. Finished Copper (µm) ($h_c$ = 105 µm) |
|---|---|---|---|---|
| Track-Track = $g_{min}$ | ≥85 | ≥100 | ≥192 | ≥250 |
| Track width = $w_{min}$ | ≥85 | ≥100 | ≥192 | ≥250 |

**Table 3.** Multilayer PCB technology available in [29].

| Layer Quantity $2n_c$ | PCB Thickness Possibility 0.80 mm | PCB Thickness Possibility 1.00 mm | PCB Thickness Possibility 1.55 mm | PCB Thickness Possibility 2.40 mm |
|---|---|---|---|---|
| 1 | Yes | Yes | Yes | Yes |
| 2 | Yes | Yes | Yes | Yes |
| 4 | Yes | Yes | Yes | Yes |
| 6 | No | Yes | Yes | Yes |
| 8 | No | No | Yes | Yes |
| 10 | No | No | Yes | Yes |

Next, the mean total length *MLT* winding length per layer is applied in the expression (16). The cupper surface area is presented in (17).

$$R_{dc} = \frac{\rho_{cu} MLT . N_c n_c}{S_{Cu}} \ [\Omega] \tag{16}$$

$$S_{Cu} = w \times h_c \tag{17}$$

where:

$$N_c n_c = N_p = \left( \frac{V_i}{4 f_s A_e B_{peak}} \right) \tag{18}$$

$$w = \frac{I_{rms}}{J_{max} h_c} = \frac{b_w - (N_c + 1)g_{min}}{N_c} \geq w_{min} \tag{19}$$

According to [30], the temperature rise in the core can be related to the exterior surface, as presented in (20), relating the thermal resistance $R_{th}$ and the total core loss. For the temperature rise in the winding, the IPC-2152 could be used. However, as shown in [31], large deviation on the winding temperature rises from IPC-2152 compared to FEA (finite element analysis) simulations can be expected. Two conclusions should be pointed out: (i) IPC-2152 is too conservative and should not be used for optimization; (ii) the maximum temperature in the core and the winding are almost the same when the losses are evenly distributed. This provides evidence that planar transformer presents elevated heat conduction between the core and the PCB winding. Therefore, keeping the same power losses on winding and core is a good practice in terms of thermal prediction.

$$\Delta T_{Core} = R_{th} P_{coreTotal} = \frac{1}{h_{ext} S_{ext}} P_{coreTotal} \tag{20}$$

### 3.2. Algorithm for Classical Transformer Design

Based on the previous section, a classical transformer design is implemented. The first step is to identify the variables present in the system. Fixed electrical variables: ($V_i$ = 20, $V_o$ = 20, $I_o$ = 2A, $I_{rms}$ = 1.1 $I_o$)

- Fixed PCB variables ($h_c$ = 105 µm, $n_c$ = 4, $H_c$ = 2 mm, g = $g_{min}$ = 192 µm, $w_{min}$ = 192 µm)
- Design manufacture variables of magnetic cores (EE $A/B/C$) are presented in Table 4.

**Table 4.** *EE model* from Ferroxcube [22].

| EE model | A [mm] | B [mm] | C [mm] |
|---|---|---|---|
| 1 | 14 | 3.5 | 5 |
| 2 | 18 | 4 | 10 |
| 3 | 22 | 6 | 16 |
| 4 | 32 | 6 | 20 |
| 5 | 38 | 10 | 25 |
| 6 | 43 | 11 | 28 |
| 7 | 58 | 11 | 38 |
| 8 | 64 | 10 | 50 |

Design variables ($\Delta T_{Core} = 50°, fs, N_c = 1,2,3 \ldots 10$)

Then, the domain, objective, and constraint functions:

- Variable domain:

$$\begin{cases} EE\ 14/3.5/5 \ldots EE\ 64/10/50 \Rightarrow 300 < V_e < 40,700 [\text{mm}^3] \\ 100 \leq f_s \leq 1000\ [\text{kHz}] \\ B_{peak} \leq B_{Sat} = 0.4\ [\text{T}] \end{cases} \tag{21}$$

- Objective function:

$$\min \left\{ \sum \left( P_{Lac} + P_{Mos} + P_{Cac} + P_{core} + P_{winding} \right) \right\} \tag{22}$$

- Constraint

$$\begin{cases} \text{a) Temperature}: R_{th} P_{coreTotal} \leq \Delta T_{CoreMax} = 50 \Rightarrow \text{This might eventually limit } N_{c_{min}} \\ \text{b) Magnetic saturation}: B_{peak} \leq B_{Sat} = 0.4\ T \Rightarrow N_{c_{min}} \geq \frac{V_i}{4 f_s A_e n_c B_{Sat}} \\ \text{c) Winding PCB}: w \geq w_{min} \Rightarrow \frac{b_w - (N_c+1)g}{N_c} \geq w_{min} \Rightarrow N_{c_{max}} \leq \frac{b_w - g_{min}}{w_{min} + g_{min}} \end{cases} \tag{23}$$

To support a fast convergence, the parameter $B_{peak}$ was expressed as a function of discrete values: number of turns $N_c$, and *Ae* from EE A/B/C, leaving only *fs* as a variable to be optimized in an outer loop containing $N_c$ iterations. From (23), it is possible to delimit from b) and c) the range of $N_c$. However, one must pay attention that such aa range can be limited by temperature constraints a), while the Joule losses will normally limit the maximum amount of $N_c$.

In addition, the algorithm intends to let the designer manage the dimensioning process, and therefore visualization and comprehension of many designs is possible rather than letting the machine produce one single result without important physical and engineering understanding, which can be essential in terms of other aspects, such as costs or potential environmental impact generation (LCA), as will be demonstrated in Section 3.1.

The results of the total power losses, and optimum *fs* could then be expressed in $N_c$ vs. *EE model*, as presented in Figures 10 and 11.

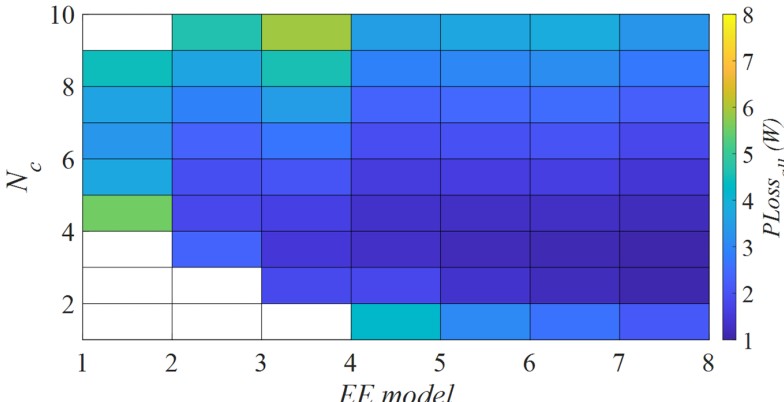

**Figure 10.** Total power losses resulted by the optimizing algorithm as function of $N_c$ vs. *EE model*.

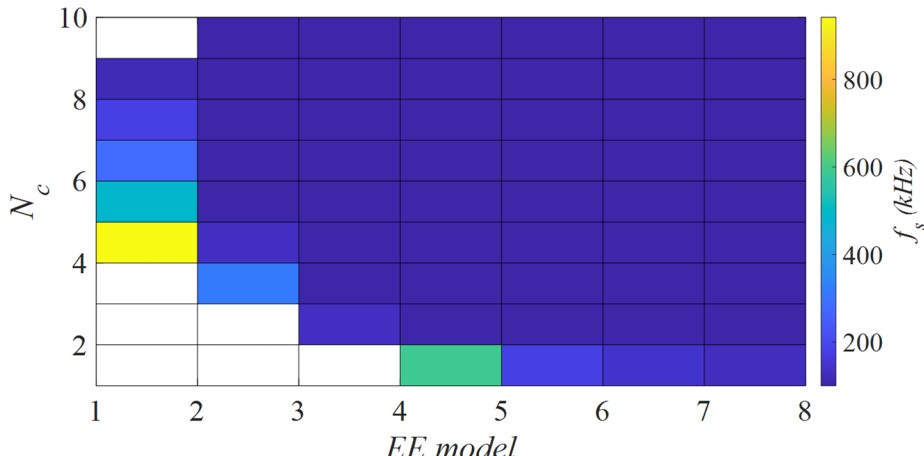

**Figure 11.** Resulting optimum *fs* as function of $N_c$ vs. *EE model*.

The regions in blank mean that no solution is possible for the constraints presented in (23), which is normal considering the small $A_e A_w$ that they present. From each *EE model*, it is possible to choose the $N_c$ that results in the minimum total power loss. It is possible to notice that, as presented in the literature, larger transformers will result in fewer power losses with the drawback of rapidly increasing volume (from 300 mm$^3$ to 40,700 mm$^3$). It also confirms that the discussion concerning efficiency versus power density is also found in a planar transformer, which goes along with the results in [32]. Based on this conclusion, the model 8 Ferroxcube E58/11/28 would be chosen in an "efficiency oriented" design. However, considering subjective Pareto analysis between efficiency and power density, it could result in smaller models.

As shown on Figure 11 the lower frequencies will mainly dominate in larger models, as the total loss is increasing with *fs*. The frequencies are therefore limited to 100 kHz. However, the inner borders will result in different frequencies due to important constraints that force the frequency to be larger than the minimum required one. The choice of PCB technology ($n_c$, $h_c$, $H_c$) makes the results vary as well.

### 3.3. Towards a Whole Life Cycle Energy and Material Impact Assessment of the Transformer

This section addresses the importance of evaluating the transformer on its whole life cycle. To perform this task, a simplified environmental impact assessment has been conducted. This streamlined approach presented aims at providing some first estimations based on a formula adapted for a transformer, to prepare the full life cycle assessment (not presented in this section). The life cycle assessment (LCA) is a method for quantifying the environmental impact of products, introduced in 1997 and framed by ISO 14040 [33]. The idea is to evaluate each process unit included in the life cycle scope of the converter to multiple environmental impacts (local and global). Global impacts include, e.g., global warming (in $CO_2$ eq.) and the ozone layer depletion, while local impacts include some local pollution in the air/water/soil (acidification, eutrophication, etc.), land use, etc. LCA tools are based on the ISO standard 14040&44 and integrate important databases that enable the aggregation of the primary flows (material, energy) included in the unit processes modeling the product life cycle (based on a functional unit, reference flows, in a defined perimeter) and the resulting potential environmental impact estimations. The environmental impact calculation is supported by several environmental impact methods (e.g., CML, Ecoindicator 99, Impact 2002+) that are discussed and improved by the worldwide LCA community.

For the perimeter of this study, the streamlined life cycle of the transformer has been divided into:

- The extraction of raw materials used for the electronic components.
- The production processes of transformer components (including the sub-systems).
- The assembly processes.

- The use stage.
- The separation process of the components; with reuse and recycling fraction end-of-life treatment scenarios (based on existing electronic waste stream states).
- The main transport stages (between each previously mentioned step).

In this study, the streamlined environmental impact assessment in this perimeter is focused on providing a first estimation of the potential embodied energy, approached through an energy cost analysis. This input flow analysis compiles the cost flow associated to the life cycle inventory to estimate the resulting embodied energy. In a line of similar reasoning to input-output flow analysis methods used in LCA, but streamlined at first, every step of the life cycle of the transformer included in the defined perimeter consumes an amount of energy, as expressed in (24). The equation is split in two terms. The term on the left is related to the energy cost of manufacturing, transport, and end of use scenarios, while the term on the right is the energy loss related to a certain given task (the energy spent during usage). This proposition allows the designers of transformer to estimate the embodied energy in regard to the technical functions and performances of the system. A cause to consequence reasoning is therefore made possible for the electronic designer. The embodied energy estimation factors may require some information exchanges with the LCA expert that has access to the full LCA software, i.e., databases (including data uncertainty) and calculations methods (including the characterization factors to estimate the environmental impacts indicator contributions, and uncertainty life cycle analysis).

$$E_{acv} = \sum M_m.E_{p,m} + \int_{use} P_{fct}dt \qquad (24)$$

The $E_{p,m}$ are the energy costs of each unit process of transformation of a given kilogram of material. The latter is obtained from LCA inventory data and databases such as Ecoinvent©, ELCD (European Reference Life Cycle Database), electronic and electrical, associated with LCA software from the BaseImpact® of the French agency ADEME, including the end-of-life current scenarios and waste stream evaluation reports, and the European Union from the EUROSTAT annual report. The $P_{fct}$ are the energy losses of the converter during its use.

The manufacturing process of the transformer is divided into three main stages plus transport:

- Extraction of the raw material.
- Manufacture of the components.
- Assembly processes.
- Transport

The end of life is divided into several scenarios including transport again:

- Recycling (material processing to obtain the same or lower quality).
- Upcycling (partial reuse of part of the converter or some components in functional condition).
- Re-use (re-use of the components depending on its condition in a new converter system).

Note that reuse and upcycling would potentially be possible on a large scale because the conversion bricks are standardized in this modular design. The conversion bricks would retain their own value for a new use. However, this re-use scenario is prospective. This model cannot be based on existing data, as it is not yet operated by the current electronic repair-reuse scheme (as in most of Europe for instance). Again, establishing some design scenario related to other life cycle stage (usage/end-of-life) supports the electronic designers' capacity to estimate (roughly) the potential embodied energy losses in its design choices.

The embodied energy of each unit modeled in the life cycle perimeter proposed in this research follows:

$$E_i = M_m.E_{unit} \qquad (25)$$

where $E_i$ is the energy per kg of a transformation process of a material stream of mass $M_m$.

The materials extracted and used in the transformer are included in Figure 12. They are all taken into account in the quantification of the energy cost of extraction of the raw material. Hence, minimizing the usage losses and the embodied energy during the whole life cycle can be expressed through the following objective function:

- Objective function:

$$\min\left\{\sum\left(P_{joule} + P_{core} + P_{winding}\right) \times years + E_{LCA}\right. \tag{26}$$

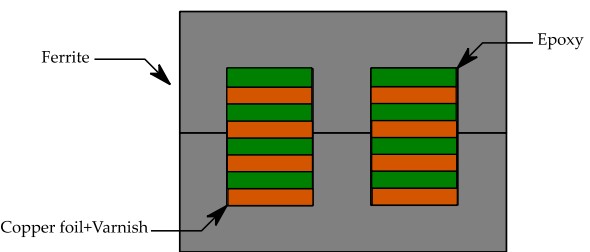

**Figure 12.** A simplified representation of planar transformer for Bill Of Material (BOM), or life cycle inventory in the streamlined perimeter of this study.

The parameter of the "streamlined LCA" is calculated over a simplified design version of the transformer (that can evolve during the design process development of the transformer to reach the best design choices compromise):

To perform the streamlined life cycle environmental impact, essentially based on an energy efficiency analysis, including the embodied energy associated to the material flow and the process included at the stages included in the study perimeter, the following parameters are taken into account, extracted from the databases presented in Table 5:

$$\begin{cases} E_{LCA} = E_{Ferrite} + E_{Copper} + E_{Epoxy+Varnish} \\ E_{Copper} = E_{Extraction+Production(FeNi)} + E_{Transport} \\ E_{Ferrite} = E_{Extraction+Production(Cu)} + E_{Transport} \\ E_{Epoxy+Varnish} = 31.4 \,\text{kWh/kg} \\ E_{Transport} = 0.6 \,\text{kWh/kg} \\ E_{Extraction+Production(Cu)} = 8.7 \,\text{kWh/kg} \\ E_{Extraction+Production(FeNi)} = 37 \,\text{kWh/kg} \\ E_{EndOfLife(WEEE)} = 31.4 \,\text{kWh/kg} \\ E_{Assembly} = 27 \,\text{kWh/kg} \end{cases}$$

**Table 5.** References for the streamlined calculation of the embodied energy: factors are extracted from calculation methods (IDEMAT datasheet, from the Idematapp 2021 cvs, cf. www.ecocostsvalue.com, and www.idematapp.com (accessed on 13 September 2021). databases used for the life cycle inventory stage (Bill of Material).

| **Raw Materials' Extraction** | **Eurostat, Ecoinvent V3.6©, [34,35]** |
| --- | --- |
| Transports | EcoTransit from the website (accessed Sept. 2021): https://www.ecotransit.org/fr (accessed on 13 September 2021), datasheet MURATA, [36,37] |
| Manufacturing | Ecoinvent V3.6© databases. |
| Usage-Maintenance | MIL-HDBK, EIME LCA software [38], MURATA database |
| Recycling and reuse | Ecoinvent©, Base impact® of the ADEME online tool (available on: www.base-impacts.ademe.fr, accessed on 13 September 2021) |

The algorithm is then run with respect to expected total converter's lifetime. Depending on components' MTBF, but also converter expected usage duration depending on application, the total lifetime can be set. For the purpose of the design optimization, a

total lifetime of ten years is selected at first. The results of the total energy losses over ten years and optimum of life cycle energy cost is expressed in $N_c$ vs. *EE model*, as presented in Figure 13.

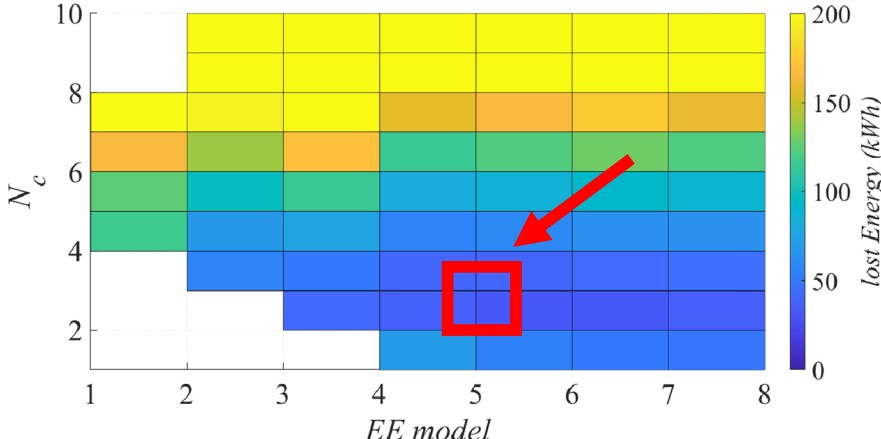

**Figure 13.** Usage energy losses and life cycle energy cost depending on the number of turns and the ferrite model.

It is noticeable that the optimum design for the transformer taking into account its whole life cycle shifts more to the left to a smaller ferrite from the Ferroxcube E58/11/28 to the E38/10/28.

An eco-efficient analysis aims to minimize the cost of energy by reducing the volume, but the electrical constraints and the design rules of the transformer push the optimum more to the right (wider transformers), which results in a possible minimum that satisfies both the energy efficiency performances and the minimization of the whole life cycle energy. The full LCA is necessary to evaluate scientifically the best life cycle design scenarios in regard to energy efficiency and other environmental impacts.

While the other indicators, such as the toxicity, ozone depletion, or land use (etc.), are important to take into account to have a global view over all the environmental impacts, it is still very difficult to implement a full LCA and more importantly to have a common indicator be easily understood by the electrical engineering community. The formulated model can be used for a first streamlined estimation in the early stage of the design process of such power electronics.

## 4. Results and Discussion

When plotting the optimized total energy lost during the usage and the embodied energy related to the manufacture, transport, and end-of-life treatments of the transformer with respect to the total lifetime, the two curves cross for a lifetime of seven years, as presented in Figure 14. Choosing a smaller transformer model E32/6/20 can therefore be a solution to reduce the energy losses, when the components around it have a short life or the whole device has a shorter life than the transformer. Nevertheless, if the device is robust and can be guaranteed to function without failure for approximately 15 years or more, then it would be interesting to over-size the transformer performance to limit the energy losses during the usage of the converter.

In our modern societies where many of our products are designed to last between five and 10 years (electric mobility such as electric bikes and scooters) or because those products will be obsolete within 2–5 years (computers, mobiles), the eco dimensioning could drive the design "naturally" toward more compact solutions, toward a higher global power density, together with a higher total energy efficiency. In this eco-design optimization, both efficiency and power densities are heading in the same direction and based on factual criteria. It is interesting to underline that the eco-design efforts could provide a complementary justification to push further power electronics designs toward

higher power densities and efficiencies, providing additional motivation for the community to keep improving both aspects in power converters.

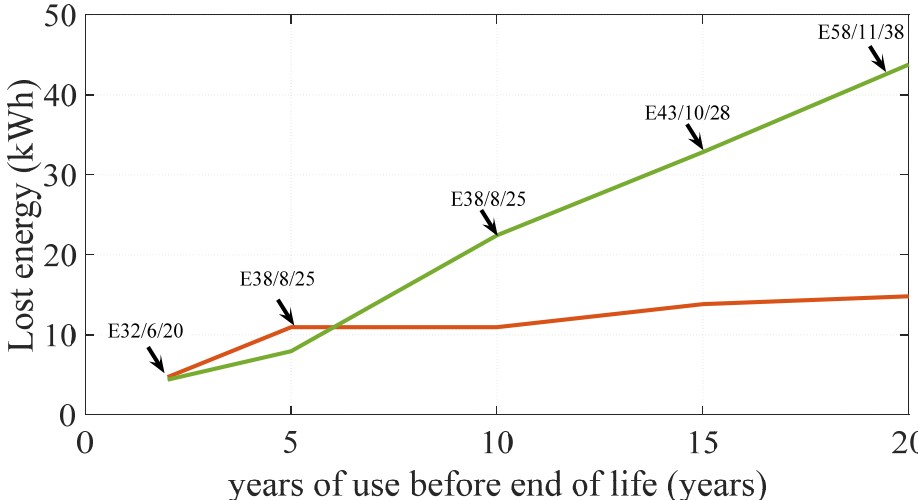

**Figure 14.** In Green the energy losses during the usage, in orange the energy losses during the other steps of the life cycle of the transformer in kWh, for example if the device has a lifetime of 5 years, the E38/8/25 Ferrite model is optimum when it comes to the whole life cycle cost.

It is important to point out here that the LCA analysis and the eco-design optimization have been partially carried out, leaving important additional work to be done taking into account the other components. Nonetheless, in DAB based converters, the transformer remains a key element to be optimized with respect to environmental issues since it represents the largest parts of the volume and weight of the whole converter. Further work will have to be carried out to check if this assumption remains valid if a full LCA analysis is done.

## 5. Conclusions

A methodology for easily designing the main magnetic devices found in a DAB with SPS modulation was proposed. This methodology allows an efficiency and power density maximization process, while ZVS is rather guaranteed for the sake of efficiency and loss of model accuracy. The transformer becomes the central key device, while other elements are designed around it.

In a conventional method, an "only" energy efficiency oriented would result in targeting the largest transformer size, while in a power density-oriented analysis the smallest transformer size would be best selected. The conventional method for determining the transformer size and the switching frequency is, therefore, most of the time subjective. On the other hand, the methodology proposed in this research allowed finding, objectively, the optimum switching frequency from the planar transformer design optimization, that can be materially and energy efficiency oriented, or even better, based on a LCA results orientation.

By following the methodology of a planar transformer for a DAB, the conventional and the proposed, environmentally (energy–material) oriented, optimized life cycle were compared. Results suggest that an increase in volume becomes interesting when the whole converter is robust enough and when its usage is guaranteed over a long period. If the case differs, the designer should choose a smaller transformer responding to the electrical constraints considered.

**Author Contributions:** Conceptualization, G.d.F.L., B.R., Y.L., J.-C.C. and M.R.; methodology, G.d.F.L., B.R., Y.L., J.-C.C. and M.R.; algorithm, G.d.F.L. and B.R.; validation, G.d.F.L. and B.R.; electrical modelling, derivation and circuit implementation, G.d.F.L., Y.L. and J.-C.C.; life cycle analysis and investigation, B.R. and M.R.; resources, G.d.F.L., B.R., Y.L., J.-C.C. and M.R.; data curation, G.d.F.L., B.R., Y.L., J.-C.C. and M.R.; writing—original draft preparation, G.d.F.L.; writing—review and editing, G.d.F.L., B.R., Y.L., J.-C.C. and M.R.; visualization, Y.L., J.-C.C. and M.R.; supervision, Y.L., J.-C.C. and M.R.; project administration, Y.L., J.-C.C. and M.R.; funding acquisition, Y.L., J.-C.C. and M.R. All authors have read and agreed to the published version of the manuscript.

**Funding:** This research received no external funding.

**Acknowledgments:** The authors would like to acknowledge Grenoble-INP and UGA for their supports and contributions selecting our research topics and providing PhD supports through IRS grants. This research work has been supported by an Auvergne Rhone Alpes FEDER funding for Mamaatec Project in partnership with MAATEL company, located in Moirans, France.

**Conflicts of Interest:** The authors declare no conflict of interest.

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
