# Peer review of "Eco-Dimensioning Approach for Planar Transformer in a Dual Active Bridge (DAB) Application"

_2673-4117, doi:10.3390/eng2040035_

Round 1

Reviewer 1 Report

The topic of the paper, evaluation of total environmental impact of power converters, is certainly important. In turn, the research is valuable, but the paper – well structured. The background of the research is well described. The problem of planar transformer eco-design is clearly formulated. So, the investigation methods, although are pure mathematical, correspond to the selected objective of reduction of environmental impact of dual active bridge converter. The obtained results are well presented. They cover the entire problem range. I recommend publishing in the current form, but have minor editorial remarks:

  • Make sure that the text in Figures 2 and 5 is readable in electronic and printed forms;
  • Also equations (11) and (13) are difficult to read;
  • At such huge number of the variables a unified list would be very useful, as well as the list of abbreviations.

Author Response

Thank you for your valuable review on our work. We hope we have addressed all your advices and corrections. Let us know if we have something to improve even more our work.

Reviewer 2 Report

Although the approach of the article shows an interesting aspect that should be taken into account in the design of electronic devices, there are many factors that influence eco-dimensioning and it is currently difficult to take them into account. The restrictions that condition electronic designs, limit the conclusions of the work carried out. On the other hand, the increasingly common use of devices e.g. SiC, more stable with temperature and with lower losses, also has a significant influence on the life of electronic equipment. Could you comment it?

Figure 14 shows the increase of power losses with time in case the of  larger magnetic core (58/11/38), but it is mentioned in line 478 that increasing its size limits the losses in it. A more in-depth explanation of this figure would help to understand the estimating energy losses with respect to the years of life.

In Fig 7 and 8, the label corresponding to the x-axis (fs (10kHz)) is not very common. To use the units directly e.g .: fs (kHz) would be preferable.

Author Response

(The authors gave the same response as above.)
